# A Humanized and Defucosylated Antibody against Podoplanin (humLpMab-23-f) Exerts Antitumor Activities in Human Lung Cancer and Glioblastoma Xenograft Models

**DOI:** 10.3390/cancers15205080

**Published:** 2023-10-20

**Authors:** Hiroyuki Suzuki, Tomokazu Ohishi, Mika K. Kaneko, Yukinari Kato

**Affiliations:** 1Department of Molecular Pharmacology, Tohoku University Graduate School of Medicine, 2-1 Seiryo-machi, Aoba-ku, Sendai 980-8575, Miyagi, Japan; k.mika@med.tohoku.ac.jp; 2Department of Antibody Drug Development, Tohoku University Graduate School of Medicine, 2-1 Seiryo-machi, Aoba-ku, Sendai 980-8575, Miyagi, Japan; 3Institute of Microbial Chemistry (BIKAKEN), Numazu, Microbial Chemistry Research Foundation, 18-24 Miyamoto, Numazu-shi 410-0301, Shizuoka, Japan; ohishit@bikaken.or.jp; 4Institute of Microbial Chemistry (BIKAKEN), Laboratory of Oncology, Microbial Chemistry Research Foundation, 3-14-23 Kamiosaki, Shinagawa-ku, Tokyo 141-0021, Japan

**Keywords:** podoplanin (PDPN), lung squamous cell carcinoma, glioblastoma, monoclonal antibody, antitumor activity, mouse xenograft model, antibody-dependent cellular cytotoxicity, complement-dependent cytotoxicity

## Abstract

**Simple Summary:**

Podoplanin (PDPN), also known as T1α/Aggrus/gp36, is a type I transmembrane sialo-glycoprotein that plays essential roles in cancer progression and metastasis. PDPN-expressing cancers show aggressive phenotypes, including increased stemness, invasiveness, and epithelial-to-mesenchymal transition, which lead to malignant progression. Furthermore, PDPN-positive cancer-associated fibroblasts mediate an immunosuppressive tumor microenvironment, which reduces antitumor immunity. Therefore, monoclonal antibodies (mAbs) against PDPN have been evaluated in preclinical models. In this study, we developed a humanized and defucosylated mAb against PDPN (humLpMab-23-f) and evaluated its antibody-dependent cellular cytotoxicity (ADCC) and antitumor effect in xenograft models of human tumor cells. humLpMab-23-f exerted antitumor activity against PDPN-overexpressed CHO-K1, endogenous PDPN-positive PC-10 (human lung squamous cell carcinoma), and LN319 (human glioblastoma) xenograft-inoculated mice.

**Abstract:**

A cancer-specific anti-PDPN mAb, LpMab-23 (mouse IgG_1_, kappa), was established in our previous study. We herein produced a humanized IgG_1_ version (humLpMab-23) and defucosylated form (humLpMab-23-f) of an anti-PDPN mAb to increase ADCC activity. humLpMab-23 recognized PDPN-overexpressed Chinese hamster ovary (CHO)-K1 (CHO/PDPN), PDPN-positive PC-10 (human lung squamous cell carcinoma), and LN319 (human glioblastoma) cells via flow cytometry. We then demonstrated that humLpMab-23-f induced ADCC and complement-dependent cytotoxicity against CHO/PDPN, PC-10, and LN319 cells in vitro and exerted high antitumor activity in mouse xenograft models, indicating that humLpMab-23-f could be useful as an antibody therapy against PDPN-positive lung squamous cell carcinomas and glioblastomas.

## 1. Introduction

Podoplanin (PDPN), also known as T1α/gp36/Aggrus, is a type I transmembrane sialo-glycoprotein that possesses an extracellular domain, transmembrane domain, and short cytoplasmic tail [1]. The PDPN extracellular domain has tandem platelet aggregation-stimulating domains, such as PLAG1, PLAG2, and PLAG3, which are associated with tumor-induced platelet aggregation [1]. A few PLAG-like domains (PLDs) have also been identified, one of which is called the PLAG4 domain [1]. In the extracellular domains of PDPN, serine or threonine residues are modified with mucin *O*-glycans with galactose β1,3 -linked to *N*-acetyl-galactosamine (GalNAc) [1]. *O*-glycosylated PLAG3 and PLAG4 have been reported to interact with C-type lectin-like receptor 2 (CLEC-2), which is important for PDPN-induced platelet aggregation [2,3]. The PDPN intracellular domain can recruit the ezrin, radixin, and moesin (ERM) complex, which regulates actin cytoskeleton reorganization and epithelial-to-mesenchymal transition (EMT) [4]. Furthermore, PDPN interacts with matrix metalloproteinases [5] and hyaluronan receptor CD44 [6]. The complex is formed in tumor invadopodia, which promotes hyaluronan binding and extracellular matrix (ECM) degradation [7]. Furthermore, PDPN mediates the diverse pattern of tumor invasion, including collective invasion in squamous cell carcinomas [8] and ameboid invasion in melanoma [9].

Overexpression of PDPN has been reported in many tumors, including squamous cell carcinomas in the head and neck, esophagus, malignant gliomas, and mesotheliomas, and is associated with poor clinical outcomes [1]. Furthermore, PDPN is also expressed in cancer-associated fibroblasts (CAFs), a major component of the tumor microenvironment (TME). CAFs are reported to enhance cancer cell survival and affect therapeutic outcomes. CAFs are also involved in the formation of an immunosuppressive TME, which can reduce antitumor immunity [10]. Elevated PDPN staining in CAFs is correlated with poor prognosis in lung [11,12,13], breast [14], and pancreatic [15] cancer patients. Therefore, PDPN has been considered as a useful marker for diagnosis and an attractive therapeutic target for tumors.

Our group previously established several mAbs against human PDPN. Clone NZ-1 exhibits neutralizing activity for CLEC-2 binding and prevents platelet aggregation and hematogenous metastasis to the lung [1]. An NZ-1-derived rat-human chimeric antibody, named NZ-8 (human IgG_1_, kappa), possesses antibody-dependent cellular cytotoxicity (ADCC) and complement-dependent cellular cytotoxicity (CDC) activities and exerts antitumor effects against malignant mesotheliomas [16]. A mouse-human chimeric anti-PDPN mAb, named chLpMab-7 (human IgG_1_, kappa), which recognizes the PLD domain, did not show neutralization activity for CLEC-2 binding but suppressed tumor growth and hematogenous metastasis to the lung, indicating that ADCC/CDC activities are essential for targeting PDPN-positive tumors [17].

To select mAbs that exhibit cancer specificity, we have established a cancer-specific monoclonal antibody (CasMab) method [18]. One CasMab, clone LpMab-2, recognizes the glycopeptide structure of PDPN from Thr55 to Leu64 [18]. In contrast, another CasMab against PDPN, clone LpMab-23, recognizes a peptide structure of PDPN from Gly54 to Leu64 [19]. We successfully applied a PDPN-targeting CasMab to chimeric antigen receptor (CAR)-T therapy in mice preclinical studies of human glioblastoma [20]. Recently, we further developed anti-HER2 CasMabs [21].

ADCC activity is mediated by natural killer (NK) cells through the interaction of FcγRIIIa with the Fc region of mAbs. The interaction between Fc and FcγRIIIa on effector cells is facilitated by a core fucose deficiency in the *N*-glycan in the Fc region [22]. Therefore, fucosyltransferase 8 (Fut8)-deficient Chinese hamster ovary (CHO) cells are the ideal hosts to produce completely defucosylated recombinant mAbs [23]. This technique was applied to the development of mogamulizumab (Poteligeo^®^), a defucosylated humanized mAb targeting CCR4 [24]. We previously produced mouse-human chimeric and defucosylated forms of anti-PDPN mAbs from LpMab-23 [19], which showed an antitumor effect in a mouse xenograft model.

In this study, we produced a humanized and defucosylated anti-PDPN mAb (humLpMab-23-f) and evaluated its ability to induce ADCC and CDC or antitumor efficacy against PDPN-positive tumor cells.

## 2. Materials and Methods

### 2.1. Cell Lines

CHO-K1 cells were obtained from the American Type Culture Collection (ATCC, Manassas, VA, USA). CHO/PDPN cells were established as previously described [18]. Human lung squamous cell carcinoma PC-10 cells were purchased from Immuno-Biological Laboratories Co., Ltd. (Gunma, Japan). CHO-K1, CHO/PDPN, and PC-10 cells were cultured in Roswell Park Memorial Institute (RPMI)-1640 medium [Nacalai Tesque, Inc. (Nacalai), Kyoto, Japan]. Human glioblastoma LN319 cells were purchased from Addexbio Technologies (San Diego, CA, USA) and cultured in Dulbecco’s Modified Eagle’s Medium (DMEM) (Nacalai). Both media were supplemented with 10% heat-inactivated fetal bovine serum [FBS; Thermo Fisher Scientific, Inc. (Thermo), Waltham, MA, USA], 100 units/mL penicillin, 100 μg/mL streptomycin, and 0.25 μg/mL amphotericin B (Nacalai). ExpiCHO-S and Fut8-deficient ExpiCHO-S (BINDS-09) cells were cultured following the manufacturer’s instructions. All cell lines were cultured at 37 °C in a humidified atmosphere with 5% CO_2_ and 95% air.

### 2.2. Recombinant mAb Production

Normal human IgG was purchased from Sigma-Aldrich Corp. (St. Louis, MO, USA). The complementarity determining region (CDR) of LpMab-23 V_H_, frame sequences of V_H_ in human Ig, and C_H_ of human IgG_1_ were cloned into the pCAG-Neo vector [FUJIFILM Wako Pure Chemical Corporation (Wako), Osaka, Japan] to generate a humanized anti-human PDPN mAb (humLpMab-23). The CDR of LpMab-23 V_L_, frame sequences of V_L_ in human Ig, and C_L_ of human kappa chain were cloned into the pCAG-Ble vector (Wako). We transfected the antibody expression vectors of humLpMab-23 into ExpiCHO-S or BINDS-09 cells using the ExpiCHO-S Expression System (Thermo Fisher Scientific, Inc.). We named these mAbs humLpMab-23 and humLpMab-23-f, respectively. We purified humLpMab-23 and humLpMab-23-f using Ab-Capcher (ProteNova Co., Ltd., Kagawa, Japan).

### 2.3. Animal Experiments for Mice Xenograft Model

To examine the antitumor effect of humLpMab-23-f, animal experiments were approved by the Institutional Committee for Experiments of the Institute of Microbial Chemistry (approval no. 2020-035). During the experimental period, we monitored mice maintained in a pathogen-free environment on 11 h light/13 h dark cycle with food and water supplied ad libitum. Mice were monitored for health and weight every one or five days. We identified body weight loss exceeding 25% and maximum tumor size exceeding 3000 mm^3^ as humane endpoints, and terminated the experiments.

### 2.4. Flow Cytometry

CHO-K1, CHO/PDPN, PC-10, and LN319 cells were collected using 0.25% trypsin and 1 mM ethylenediamine tetraacetic acid (EDTA; Nacalai). The cells (1 × 10^5^ cells/sample) were treated with humLpMab-23 or blocking buffer (control) (0.1% BSA in PBS) for 30 min at 4 °C. Next, the cells were treated with fluorescein isothiocyanate (FITC)-conjugated anti-human IgG (1:2000; Sigma-Aldrich Corp., St. Louis, MO, USA) for 30 min at 4 °C. The EC 800 Cell Analyzer (Sony Corp., Tokyo, Japan) was used to collect the fluorescence data, which were analyzed using FlowJo software [BD Biosciences (BD), Franklin Lakes, NJ, USA].

### 2.5. Determination of the Binding Affinity by Flow Cytometry

After being suspended in 100 μL of serially diluted humLpMab-23, the cells were then incubated with FITC-conjugated anti-human IgG (1:200). The EC800 Cell Analyzer was then used to gather fluorescence data. The binding isotherms were fitted into the built-in, one-site binding model using GraphPad PRISM 8 software (GraphPad Software, Inc., La Jolla, CA, USA) to calculate the dissociation constant (*K*_D_).

### 2.6. ADCC

We purchased human NK cells from Takara Bio, Inc. (Shiga, Japan). We labeled target cells (CHO/PDPN, PC-10, and LN319) using 10 µg/mL Calcein AM (Thermo Fisher Scientific, Inc.) and resuspended them. Target cells were plated in 96-well plates (1 × 10^4^ cells/well) and mixed with the human NK cells and 100 μg/mL of control human IgG or humLpMab-23-f. The calcein released to the supernatant from each well was measured after a 4.5 h incubation. Fluorescence intensity was determined using a microplate reader (Power Scan HT; BioTek Instruments, Winooski, VT, USA) with excitation and emission wave lengths of 485 and 538 nm, respectively. After lysing all cells with a buffer containing 0.5% Triton X-100, 10 mM Tris-HCl (pH 7.4), and 10 mM EDTA, cytotoxicity (% lysis) was calculated as % lysis = (E − S)/(M − S) × 100, where E is the fluorescence of the combined target and effector cells, S is the spontaneous fluorescence of target cells only, and M is the maximum fluorescence measured.

### 2.7. CDC

The calcein AM-labeled target cells (CHO/PDPN, PC-10, and LN319) were plated and mixed with rabbit complement (final dilution 1:10, Low-Tox-M Rabbit Complement; Cedarlane Laboratories, Hornby, ON, Canada) and 100 μg/mL of control human IgG or humLpMab-23-f. Following incubation for 4.5 h at 37 °C, the calcein released into the medium was measured, as described above.

### 2.8. Antitumor Activity of humLpMab-23-f in Xenografts of CHO-K1, CHO/PDPN, PC-10, and LN319 Cells

First, each cell was suspended in 0.3 mL of 1.33 × 10^8^ cells/mL using DMEM and mixed with 0.5 mL of BD Matrigel Matrix Growth Factor Reduced (BD Biosciences, San Jose, CA, USA). Then, BALB/c nude mice (Jackson Laboratory Japan, Kanagawa, Japan) were injected subcutaneously in the left flank with 100 μL of the suspension (5 × 10^6^ cells). After day six, we injected the mice with 100 μg of humLpMab-23-f (*n* = 8) and control human IgG (*n* = 8) in 100 μL of PBS through intraperitoneal injection (i.p.). Additional antibodies were injected on days 14 and 21.

Human NK cells (8.0 × 10^5^ cells) were injected around the tumors on days 6, 14, and 21. The tumor diameter was measured on days 6, 11, 14, 18, 21, and 25 after inoculation with cells. The tumor volume was calculated using the following formula: volume = W^2^ × L/2, where W is the short diameter and L is the long diameter. All mice were euthanized by cervical dislocation 25 days after cell inoculation.

### 2.9. Statistical Analyses

All data are represented as the mean ± standard error of the mean (SEM). Welch’s *t*-test was used for the statistical analyses of ADCC, CDC, and tumor weight. ANOVA with Sidak’s post hoc test was used for tumor volume and mouse weight. *p* < 0.05 was considered to indicate a statistically significant difference.

## 3. Results

### 3.1. Production of Humanized Anti-PDPN mAb, humLpMab-23

We previously established an anti-PDPN mAb (LpMab-23; mouse IgG_1_, kappa) by immunization with the PDPN ectodomain produced by glioblastoma LN229 cells [19]. LpMab-23 was shown to be useful for flow cytometry [19]. In this study, we engineered a humanized LpMab-23 (humLpMab-23) by fusing the V*_H_* and V*_L_* CDRs of LpMab-23 with the C*_H_* and C*_L_* chains of human IgG_1_, respectively (Figure 1A). humLpMab-23 was detected in CHO/PDPN cells, but not in parental CHO-K1 cells (Figure 1B). Furthermore, humLpMab-23 reacted with PDPN-positive glioblastoma LN319 and lung squamous cell carcinoma PC-10 cells (Figure 1C).

The *K*_D_ for the interaction of humLpMab-23 with CHO/PDPN, LN319, and PC-10 cells was determined by flow cytometry. As shown in Figure 1D, the *K*_D_ values for humLpMab-23 with CHO/PDPN, LN319, and PC-10 cells were 4.7 × 10^−9^ M, 4.9 × 10^−9^ M, and 5.4 × 10^−9^ M, respectively. These results suggested that humLpMab-23-f demonstrated a high affinity for PDPN-positive cells.

### 3.2. ADCC and CDC by Humlpmab-23-f against CHO/PDPN Cells

Next, we developed the core fucose-deficient version of humLpMab-23, named humLpMab-23-f, using BINDS-09 cells (Figure 1A). We then examined if humLpMab-23-f showed ADCC activity against CHO/PDPN cells. As shown in Figure 2A, humLpMab-23-f induced ADCC against CHO/PDPN cells (51.6% cytotoxicity) more effectively than the control human IgG (13.6% cytotoxicity; *p* < 0.05). No difference was observed in ADCC against CHO-K1 cells between the humLpMab-23-f and control human IgG-treated groups (Figure 2B).

Furthermore, we investigated CDC by humLpMab-23-f against CHO/PDPN cells. As shown in Figure 2C, humLpMab-23-f induced greater CDC against CHO/PDPN cells (69.2% cytotoxicity) compared with that induced by control human IgG (15.0% cytotoxicity; *p* < 0.05). No difference was observed in CDC against CHO-K1 cells between the humLpMab-23-f and control human IgG-treated groups (Figure 2D).

These results demonstrated that humLpMab-23-f exhibited potent ADCC and CDC activities against CHO/PDPN cells.

### 3.3. Antitumor Effects of Humlpmab-23-f against CHO/PDPN Xenografts

Intraperitoneally, humLpMab-23-f and control human IgG were administered on days 6, 14, and 21, following inoculation with CHO/PDPN cells in BALB/c nude mice. Furthermore, human NK cells were injected around the tumors on the same days as the mAb injection. The tumor volume was measured on days 6, 11, 14, 18, 21, and 25 after inoculation with CHO/PDPN cells. humLpMab-23-f administration resulted in a significant reduction in tumor volume on days 18 (*p* < 0.01), 21 (*p* < 0.01), and 25 (*p* < 0.01) compared with that induced by the human IgG (Figure 3A). humLpMab-23-f treatment resulted in about 73% reduction in volume compared with that induced by the control human IgG on day 25 post-injection. When treated with humLpMab-23-f, the weight of CHO/PDPN tumors was significantly lower compared with those treated with control human IgG (90% reduction; *p* < 0.01; Figure 3C). Figure 3E demonstrates the CHO/PDPN tumors resected from mice on day 25. Body weight was not affected in CHO/PDPN tumor-bearing mice treated with humLpMab-23-f and control human IgG (Figure 3G).

In the CHO-K1-inoculated mice, humLpMab-23-f and control human IgG were injected intraperitoneally into mice on days 6, 14, and 21 after inoculation. Furthermore, human NK cells were injected around the tumors on the same days as the mAb injection. There was no difference in CHO-K1 xenograft volume (Figure 3B) and weight (Figure 3D) between the humLpMab-23-f and control human IgG-treated groups. Figure 3F demonstrates the CHO-K1 tumors that were resected from mice on day 25. Body weight loss was not observed (Figure 3H). Appendix A presents the body appearance of CHO/PDPN and CHO-K1 xenograft-inoculated mice treated with human IgG and humLpMab-23-f on day 25.

### 3.4. ADCC and CDC by Humlpmab-23-f against LN319 and PC-10 Cells

We next investigated ADCC by humLpMab-23-f against endogenous PDPN-positive LN319 and PC-10 cells. As shown in Figure 4A, humLpMab-23-f showed more potent ADCC against LN319 cells (47.7% cytotoxicity) than that induced by control human IgG (15.8% cytotoxicity; *p* < 0.01). Furthermore, humLpMab-23-f showed more potent ADCC against PC-10 cells (46.7% cytotoxicity) than that induced by control human IgG (14.6% cytotoxicity; *p* < 0.01) (Figure 4B).

For CDC by humLpMab-23-f against LN319 cells, humLpMab-23-f exhibited greater CDC (34.7% cytotoxicity) compared with that induced by control human IgG (11.6% cytotoxicity; *p* < 0.05) (Figure 4C). Moreover, humLpMab-23-f exhibited greater CDC against PC-10 cells (53.4% cytotoxicity) compared with that induced by control human IgG (15.0% cytotoxicity; *p* < 0.01) (Figure 4D).

These results demonstrated that humLpMab-23-f exhibited potent ADCC and CDC activities against endogenous PDPN-positive LN319 and PC-10 cells.

### 3.5. Antitumor Effects of Humlpmab-23-f in LN319 and PC-10 Xenografts

In the LN319 and PC-10 xenograft tumor-bearing mice, humLpMab-23-f and control human IgG were intraperitoneally administered on days 6, 14, and 21. Furthermore, human NK cells were injected around the tumors on the same days as mAb administration. The tumor volume was measured on days 6, 11, 14, 18, 21, and 25 after inoculation. humLpMab-23-f treatment significantly reduced the volume of LN319 xenografts on days 21 (*p* < 0.01) and 25 (*p* < 0.01) compared with that induced by the control human IgG (Figure 5A). humLpMab-23-f also induced a significant reduction in PC-10 xenograft volume on days 18 (*p* < 0.01), 21 (*p* < 0.01), and 25 (*p* < 0.01) compared with that induced by the control human IgG (Figure 5B). humLpMab-23-f treatment resulted in 64% (LN319) and 69% (PC-10) decrease in tumor weight compared with that induced by the control human IgG on day 25 (Figure 5C,D). Figure 5E,F demonstrates the LN319 and PC-10 tumors that were resected on day 25.

Body weight loss was not observed in either LN319 or PC-10 xenograft-bearing mice treated with humLpMab-23-f and control human IgG (Figure 5G,H). Appendix A presents the body appearance of LN319 and PC-10 xenograft-inoculated mice treated with human IgG and humLpMab-23-f on day 25.

## 4. Discussion

In this study, we showed that humLpMab-23-f exhibited potent ADCC and CDC activities against CHO/PDPN, PC-10, and LN319 cells (Figure 2 and Figure 4). Furthermore, administration of humLpMab-23-f in tumor-bearing mice completely suppressed the growth of CHO/PDPN xenografts (Figure 3A,C,E) and exerted potent antitumor effects in PC-10 and LN319 xenografts (Figure 5). These results indicated that humLpMab-23-f could be useful as an antibody therapy for PDPN-positive human tumors.

We previously evaluated defucosylated mouse-human chimeric LpMab-23 (chLpMab-23-f) [19]. Although chLpMab-23-f exhibited ADCC activity and exerted a moderate antitumor effect against CHO/PDPN cells, chLpMab-23-f neither exhibited ADCC activity against PC-10 cells nor CDC activity against PDPN-expressing tumor cells [19]. The results of the binding affinity experiments revealed that the *K*_D_ of humLpMab-23 with PC-10 cells was 5.4 × 10^−9^ M (Figure 1D). In contrast, the *K*_D_ of chLpMab-23 with PC-10 cells was 1.4 × 10^−7^ M (Appendix A). These results indicated that humLpMab-23 shows a superior binding affinity compared to chLpMab-23. The increased binding affinity is thought to be a reason why humLpMab-23-f exerted more potent antitumor effects than chLpMab-23.

PDPN is involved in tumor malignant progression through activation of tumor invasiveness, EMT, and stemness. Furthermore, PDPN induces platelet aggregation through interaction with CLEC-2 on platelets, which is associated with venous thromboembolism and evasion of immune cells [25,26]. Therefore, several anti-PDPN mAbs have been evaluated in preclinical studies. Our group demonstrated that chLpMab-7 (a mouse-human chimeric anti-PDPN mAb) and NZ-8 (a rat-human chimeric antibody derived from NZ-1) exerted antitumor effects through their ADCC and CDC activities. Furthermore, NZ-1-R700 conjugate [27] and NZ-16 labeled with actinium-225 (^225^Ac) [28] have been developed as near-infrared photoimmunotherapy and radioimmunotherapy agents against PDPN, respectively. Both mAbs also exhibited antitumor effects against the malignant pleural mesothelioma xenograft model. In addition, neutralizing mAbs for PDPN-CLEC-2 interaction suppressed hematogenous pulmonary metastasis by inhibiting platelet aggregation [29]. LpMab-23 has a different epitope from the abovementioned anti-PDPN mAbs [19] and showed comparable ADCC/CDC activities and antitumor effects in vivo (Figure 2, Figure 3, Figure 4 and Figure 5). In contrast, the reactivity of LpMab-23 to normal cells such as lymphatic endothelial cells was low in immunohistochemical analysis [30]. It should be determined whether humLpMab-23-f induces reduced cytotoxicity against normal PDPN-expressing cells. We already confirmed that mouse-human chimeric LpMab-23 (20 mg/kg) [19] did not show any toxicity against cynomolgus monkeys.

CAFs are important players in the TME. CAFs are involved in tumor progression through the promotion of tumor proliferation, invasion, and chemoresistance, as well as ECM remodeling and immunosuppression [31]. PDPN expression is also observed in CAFs. Increased expression of PDPN in CAFs from lung [11,12,13], pancreas [15], and breast [14] tumors is associated with poor prognosis of patients. In PDPN-positive CAFs from lung tumors, the CAFs promote lung cancer cell resistance to inhibitors of EGFR [32]. Furthermore, increased expression of TGF-β was observed in PDPN-positive CAFs [13], which are associated with an immunosuppressive TME [10]. In breast cancer, the S100A4/PDPN expression ratio in CAFs is associated with disease outcomes across subtypes. High PDPN levels in CAFs were significantly correlated with shorter recurrence-free survival and overall survival [33]. Moreover, PDPN-positive CAFs were associated with low IL-2 activity and resistance to trastuzumab in HER2-positive breast cancer patients [34]. These findings indicate that the character of CAFs is an important predictive marker of breast cancer progression and resistance to treatment. Therefore, targeting PDPN-positive CAFs using anti-PDPN mAbs could be a strategy for tumor therapy. Since we have not examined the reactivity of humLpMab-23 to CAFs, further studies to characterize humLpMab-23-reactive CAFs are needed.

PD-1 and CTLA-4 on effector T lymphocytes play essential functions in tumor immune response. PDPN was reported as a co-inhibitory receptor on both CD4^+^ and CD8^+^ T lymphocytes. In T lymphocyte-specific *Pdpn*-deleted mice, the growth of inoculated B16F10 tumors was significantly delayed. The PDPN-deficient CD8^+^ tumor-infiltrating lymphocytes exhibited increased tumor necrosis factor production, while the abundance of exhausted T lymphocytes was decreased [35]. Moreover, deletion of *Pdpn* in T cells resulted in the development of autoimmune encephalomyelitis with increased accumulation of effector CD4^+^ T lymphocytes [36]. These results suggest the involvement of PDPN in tumor immune suppression. Although the ligand of PDPN involved in suppressing T lymphocytes has not been identified, a PDPN-targeting mAb has potential as a novel immune checkpoint inhibitor. Therefore, the combinatorial effects of anti-PDPN mAbs with immune checkpoint inhibitors should be considered in preclinical studies. We have established anti-PDPN mAbs against various species, including mice [1]. The anti-mouse PDPN mAbs could be useful in revealing the importance of PDPN-positive lymphocytes in cancer immunity and obtaining proof-of-concept for tumor therapy.

## Figures and Tables

**Figure 1 cancers-15-05080-f001:**
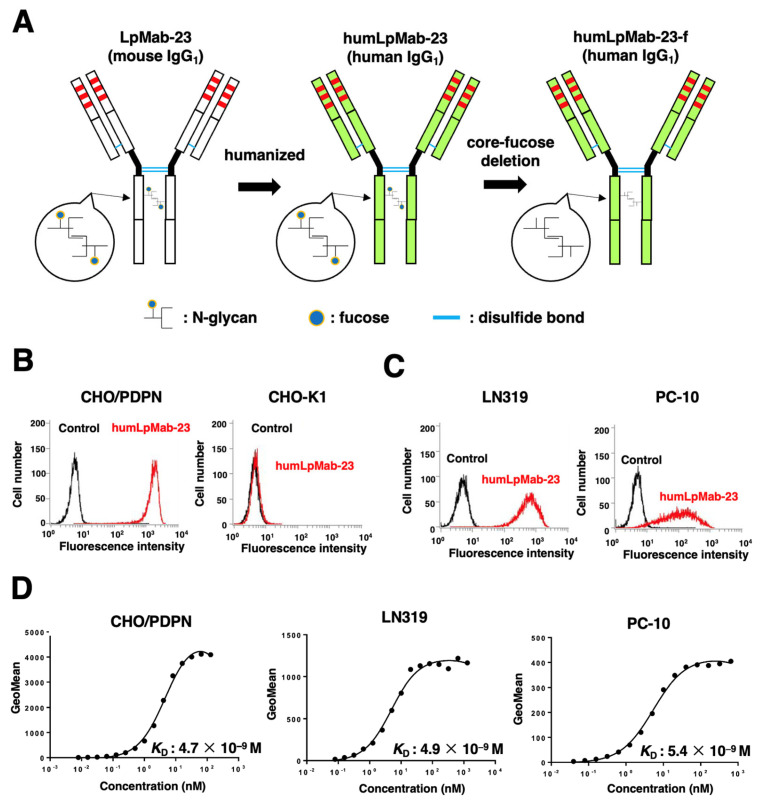
Flow cytometry using humLpMab-23. (**A**) A humanized IgG_1_ mAb, humLpMab-23, was generated from LpMab-23-f (mouse IgG_1_). The core fucose-deficient form (humLpMab-23-f) was produced using Fut8-knockout ExpiCHO-S cells. (**B**) CHO/PDPN and CHO-K1 cells were treated with humLpMab-23 (1 µg/mL) or buffer control, followed by anti-human IgG conjugated with FITC. (**C**) LN319 and PC-10 cells were treated with humLpMab-23 (1 µg/mL) or buffer control, followed by Alexa Fluor 488-conjugated anti-human IgG. (**D**) Determination of the binding affinity of humLpMab-23 using flow cytometry. CHO/PDPN, LN319, and PC-10 cells were suspended in humLpMab-23 at indicated concentrations, followed by treatment with anti-human IgG conjugated with FITC. The EC800 Cell Analyzer was used to analyze fluorescence data. GraphPad Prism 8 was used to determine the dissociation constant (*K*_D_).

**Figure 2 cancers-15-05080-f002:**
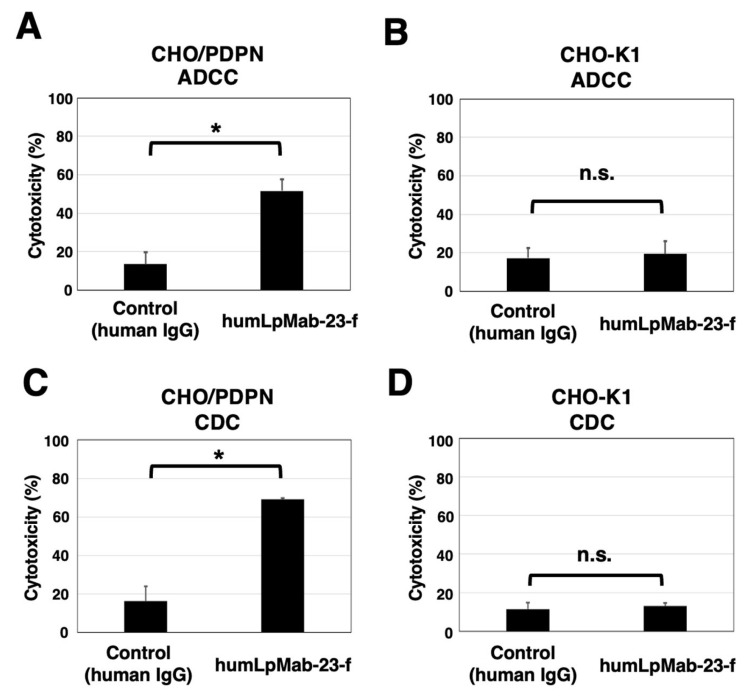
The ADCC and CDC activities mediated by humLpMab-23-f in CHO-K1 and CHO/PDPN cells. (**A**,**B**) The ADCC induced by humLpMab-23-f or control human IgG against CHO/PDPN (**A**) and CHO-K1 (**B**) cells. (**C**,**D**) The CDC induced by humLpMab-23-f or control human IgG against CHO/PDPN (**C**) and CHO-K1 (**D**) cells. Values are shown as the mean ± SEM. Asterisks indicate statistical significance (* *p* < 0.05; Welch’s *t*-test). n.s., not significant.

**Figure 3 cancers-15-05080-f003:**
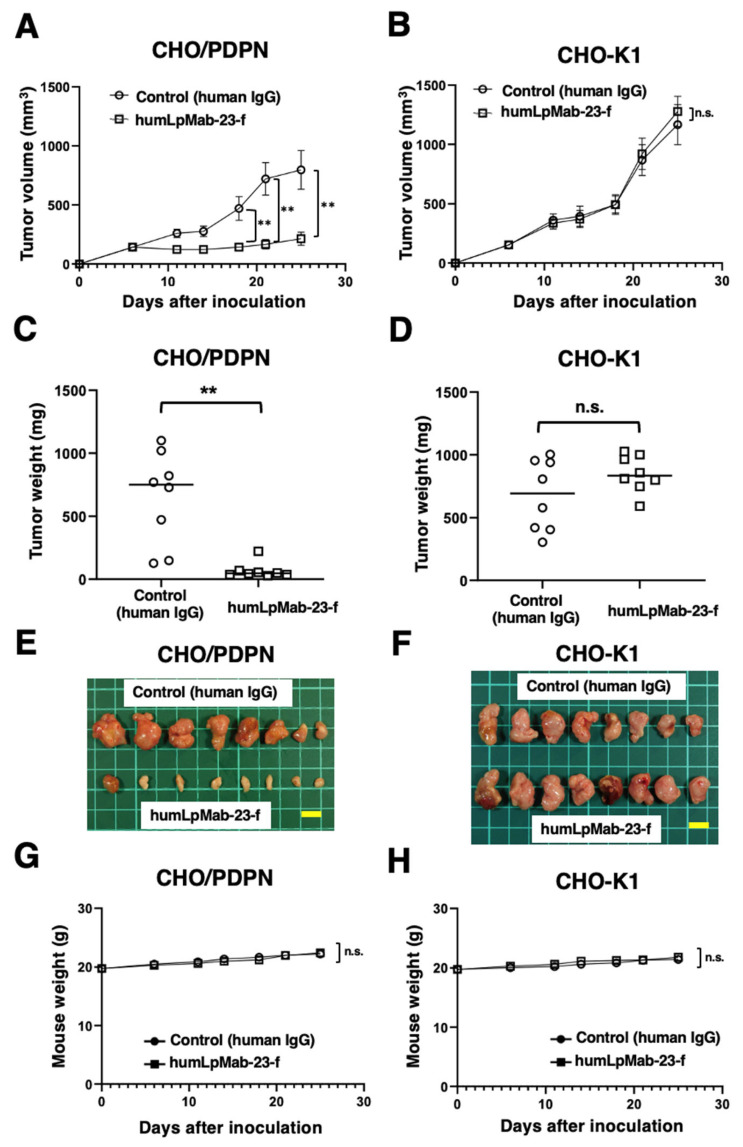
The antitumor effect of humLpMab-23-f against CHO-K1 and CHO/PDPN xenografts. (**A**,**B**) CHO/PDPN (**A**) and CHO-K1 (**B**) cells were subcutaneously inoculated into BALB/c nude mice (day 0). On day 6, 100 μg of humLpMab-23-f or control human IgG was administered intraperitoneally. Additional antibodies were administered on days 14 and 21. Human NK cells were injected around the tumors on the same days as mAb administration. Tumor volume was measured on days 6, 11, 14, 18, 21, and 25. Values are presented as the mean ± SEM. ** *p* < 0.01 (ANOVA and Sidak’s multiple comparisons test). (**C**,**D**) Weight of CHO/PDPN (**C**) and CHO-K1 (**D**) xenograft tumors on day 25. Values are represented as the mean ± SEM. ** *p* < 0.01 (Welch’s *t*-test). (**E**,**F**) The CHO/PDPN (**E**) and CHO-K1 (**F**) xenograft tumors on day 25 (scale bar, 1 cm). (**G**,**H**) Body weight of CHO/PDPN (**G**) and CHO-K1 (**H**) xenograft-bearing mice treated with control human IgG and humLpMab-23-f. n.s., not significant.

**Figure 4 cancers-15-05080-f004:**
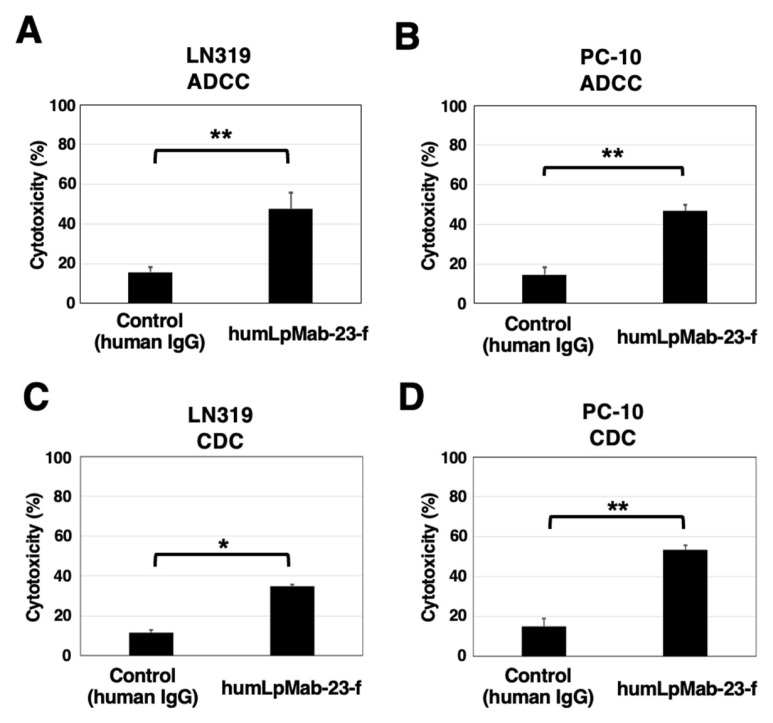
The ADCC and CDC activities mediated by humLpMab-23-f in PDPN-positive LN319 and PC-10 cells. (**A**,**B**) ADCC induced by humLpMab-23-f or control human IgG against LN319 (**A**) and PC-10 (**B**) cells. (**C**,**D**) CDC induced by humLpMab-23-f or control human IgG against LN319 (**C**) and PC-10 (**D**) cells. Values are shown as the mean ± SEM. Asterisks indicate statistical significance (* *p* < 0.05, ** *p* < 0.01; Welch’s *t*-test).

**Figure 5 cancers-15-05080-f005:**
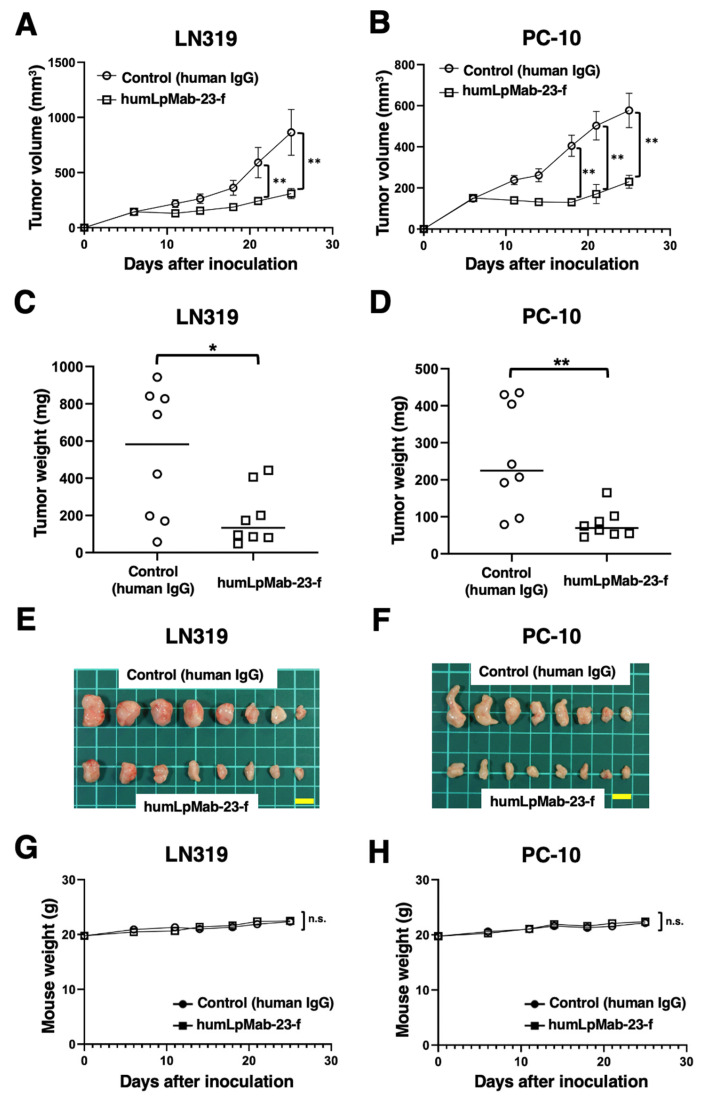
Antitumor activity of humLpMab-23-f against LN319 and PC-10 xenografts. (**A**,**B**) LN319 (**A**) and PC-10 (**B**) cells were subcutaneously injected into BALB/c nude mice (day 0). On day 6, 100 μg of humLpMab-23-f or control human IgG was administered. Additional antibodies were administered on days 14 and 21. Human NK cells were injected around the tumors on the same days as mAb administration. The tumor volume was measured on days 6, 11, 14, 18, 21, and 25. Values are presented as the mean ± SEM. ** *p* < 0.01 (ANOVA and Sidak’s multiple comparisons test). (**C**,**D**) Weight of LN319 (**C**) and PC-10 (**D**) xenograft tumors on day 25. Values are represented as the mean ± SEM. * *p* < 0.05; ** *p* < 0.01 (Welch’s *t*-test). (**E**,**F**) The LN319 (**E**) and PC-10 (**F**) xenograft tumors on day 25 (scale bar, 1 cm). (**G**,**H**) Body weight of LN319 (**G**) and PC-10 (**H**) xenograft-bearing mice treated with control human IgG and humLpMab-23-f. n.s., not significant.

## Data Availability

The data presented in this study are available in the article and Appendix A.

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
