# Peer review of "A Humanized and Defucosylated Antibody against Podoplanin (humLpMab-23-f) Exerts Antitumor Activities in Human Lung Cancer and Glioblastoma Xenograft Models"

_cancers, 2023, doi:10.3390/cancers15205080_

Round 1
Reviewer 1 Report
This study looks to develop a humanized and defucosylated form of a therapeutic anti-podoplanin antibody that the authors have previously developed.
They present an extensive and thorough examination of the new antibody form and demonstrate impressive anti-tumour activity in a range of engineered and naturally-expressing subcutaneous xenograft models. Whilst it would have improved the study to see efficacy in more complex orthotopic or metastatic tumour models, the initial results as presented show a lot of promise for this therapeutic approach.
Apart from issues with the writing and structure (commented on below), the main issue is that it is not until the results section that the tumour types for the cell lines are revealed earlier in the manuscript. Also for the animal work, for clarity, full details of husbandry should be given in the manuscript rather than being sent to an older reference. In addition the conclusions could be stronger with clearer coverage of the advantages and benefits of the humanised and defucosylated version.
Some of the sentence construction is poor in places, and the presentation flow could be better.
Author Response
This study looks to develop a humanized and defucosylated form of a therapeutic anti-podoplanin antibody that the authors have previously developed.
They present an extensive and thorough examination of the new antibody form and demonstrate impressive anti-tumour activity in a range of engineered and naturally-expressing subcutaneous xenograft models. Whilst it would have improved the study to see efficacy in more complex orthotopic or metastatic tumour models, the initial results as presented show a lot of promise for this therapeutic approach.
Apart from issues with the writing and structure (commented on below), the main issue is that it is not until the results section that the tumour types for the cell lines are revealed earlier in the manuscript.
We added the information of cell lines in “Simple Summary” and “Materials and Methods”.
Also for the animal work, for clarity, full details of husbandry should be given in the manuscript rather than being sent to an older reference.
We added the information of animal experiments in detail.
In addition the conclusions could be stronger with clearer coverage of the advantages and benefits of the humanised and defucosylated version.
We already described an advantage of humanized version in discussion, as follows.
“The results of binding affinity revealed that KD of humLpMab-23 against PC-10 was 5.4 × 10−9 M (Figure 1D). In contrast, KD of chLpMab-23 against PC-10 was 1.4 × 10−7 M (Supplemental Figure S3). These results indicated that humLpMab-23 shows a superior binding affinity compared to chLpMab-23.”
We have not compared the difference between normal and defucosylated version of humLpMab-23. Therefore, we added the information of clinically approved defucosylated mAb, mogamulizumab (Poteligeo) which is produced by the same method of this study.
Reviewer 2 Report
Specific comments to the authors
The authors Hiroyuki Suzuki et al. of the submitted manuscript “Humanized and defucosylated antibody against Podoplanin (humLpMab-23-f) exerted antitumor activities in human tumor xenograft models” studie the anti-cancer possibilities of self developed monoclonal antibodies (mAbs) against podoplanin in preclinical in-vitro and in-vivo models.
In summary, the authors were able to show that (i) humLpMab-23 recognised PDPN-overexpressing cell lines by flow cytometry. (ii) the designed humLpMab-23-f exerted ADCC and CDC cytotoxicity against CHO/PDPN, PC-10 and LN319 cells in vitro (iii) with potent antitumour activities in the applied xenograft models. Therefore, the authors postulate that humLpMab-23-f may be useful for an antibody treatment regimen for PDPN-positive human cancers in the future.
In summary, the submitted manuscript provides some interesting information and insights into the generation of proprietary monoclonal antibodies (mAbs) against podoplanin in preclinical models, apparently via ADCC and CDC related mechanisms. The manuscript (including the presentation) is clear and convincing. The methods are mostly well described. Although the results and discussion are clearly presented, the authors (see specific comments) need to make some minor to major changes to improve the manuscript. In conclusion, the data presented are interesting. After incorporating the specific comments mentioned (see below), the manuscript has the potential to be accepted.
Specific comments
Title: Please specify which tumour cell lines will be used.
Introduction: Please specify any abbreviations such as ADCC and CDC before use.
Materials and methods: Please specify the tumour entities of the cell lines used.
Results:
# Figure 3 and 5: Regarding the demonstrated ADCC and DCC mediated effects of humLpMab-23-f (see Figure 2 and 4), further histochemical and immunohistochemical analyses of the treated xenografts are needed to demonstrate these effects in vivo. Further laboratory and histological analyses of kidney, liver and bone marrow are needed to exclude organ-specific toxicity by the application of humLpMab-23-f.
Discussion: Regarding the mentioned influence of podoplanin on EMT and CAFs, the authors should show/demonstrate these effects of their designed antibody in the xenograft model. Furthermore, the combinatorial effects of humLpMab-23-f and checkpoint inhibitors should also be investigated. Finally, how could the interesting findings be translated from a theoretical to a practical point of view (e.g. clinical setting (drug development, drug combination))? Please discuss briefly.
Minor editing of English language required
Author Response
Specific comments to the authors
The authors Hiroyuki Suzuki et al. of the submitted manuscript “Humanized and defucosylated antibody against Podoplanin (humLpMab-23-f) exerted antitumor activities in human tumor xenograft models” studie the anti-cancer possibilities of self developed monoclonal antibodies (mAbs) against podoplanin in preclinical in-vitro and in-vivo models.
In summary, the authors were able to show that (i) humLpMab-23 recognised PDPN-overexpressing cell lines by flow cytometry. (ii) the designed humLpMab-23-f exerted ADCC and CDC cytotoxicity against CHO/PDPN, PC-10 and LN319 cells in vitro (iii) with potent antitumour activities in the applied xenograft models. Therefore, the authors postulate that humLpMab-23-f may be useful for an antibody treatment regimen for PDPN-positive human cancers in the future.
In summary, the submitted manuscript provides some interesting information and insights into the generation of proprietary monoclonal antibodies (mAbs) against podoplanin in preclinical models, apparently via ADCC and CDC related mechanisms. The manuscript (including the presentation) is clear and convincing. The methods are mostly well described. Although the results and discussion are clearly presented, the authors (see specific comments) need to make some minor to major changes to improve the manuscript. In conclusion, the data presented are interesting. After incorporating the specific comments mentioned (see below), the manuscript has the potential to be accepted.
Specific comments
Title: Please specify which tumour cell lines will be used.
We added.
Introduction: Please specify any abbreviations such as ADCC and CDC before use.
We added.
Materials and methods: Please specify the tumour entities of the cell lines used.
We added.
Results:
# Figure 3 and 5: Regarding the demonstrated ADCC and DCC mediated effects of humLpMab-23-f (see Figure 2 and 4), further histochemical and immunohistochemical analyses of the treated xenografts are needed to demonstrate these effects in vivo. Further laboratory and histological analyses of kidney, liver and bone marrow are needed to exclude organ-specific toxicity by the application of humLpMab-23-f.
We previously demonstrated that mouse-human chimeric LpMab-23 (20 mg/kg) did not show any toxicities against cynomolgus monkeys. In contrast, LpMab-23 does not cross-react with mouse podoplanin; therefore, we do not need to check its toxicity against mouse xenograft models.
Discussion: Regarding the mentioned influence of podoplanin on EMT and CAFs, the authors should show/demonstrate these effects of their designed antibody in the xenograft model. Furthermore, the combinatorial effects of humLpMab-23-f and checkpoint inhibitors should also be investigated. Finally, how could the interesting findings be translated from a theoretical to a practical point of view (e.g. clinical setting (drug development, drug combination))? Please discuss briefly.
We added brief discussions from line 422~423 and 434~438.
Reviewer 3 Report
No questions.
Author Response
Thank you very much